# MILDSum: A Novel Benchmark Dataset for Multilingual Summarization of Indian Legal Case Judgments

**Debtanu Datta**[*], **Shubham Soni**[*], **Rajdeep Mukherjee, Saptarshi Ghosh**
Indian Institute of Technology Kharagpur
debtanumathcs@kgpian.iitkgp.ac.in, shubhamsonikgp@gmail.com,
rajdeep1989@iitkgp.ac.in, saptarshi@cse.iitkgp.ac.in

## Abstract

Automatic summarization of legal case judgments is a practically important problem that has attracted substantial research efforts in many countries. In the context of the Indian judiciary, there is an additional complexity – Indian legal case judgments are mostly written in complex English, but a significant portion of India's population lacks command of the English language. Hence, it is crucial to summarize the legal documents in Indian languages to ensure equitable access to justice. While prior research primarily focuses on summarizing legal case judgments in their source languages, this study presents a pioneering effort toward cross-lingual summarization of English legal documents into Hindi, the most frequently spoken Indian language. We construct the first high-quality legal corpus comprising of 3,122 case judgments from prominent Indian courts in English, along with their summaries in both English and Hindi, drafted by legal practitioners. We benchmark the performance of several diverse summarization approaches on our corpus and demonstrate the need for further research in cross-lingual summarization in the legal domain.

## 1 Introduction

Legal judgment summarization is an important and challenging task, especially considering the lengthy and complex nature of case judgments (Shukla et al., 2022; Bhattacharya et al., 2019; Aumiller et al., 2022). In the context of the Indian judiciary, there is an additional requirement – Indian legal case judgments are mostly written in complex English due to historical reasons, but a significant portion of India's population lacks a strong command of the English language. Hence, it is important to summarize case judgements in Indian languages to ensure equitable access to justice.

There exist a few Indian legal case judgment summarization datasets, e.g., the datasets developed by Shukla et al. (2022), but all of them contain English case documents and summaries only. In this work, we introduce **MILDSum** (**M**ultilingual **I**ndian **L**egal **D**ocument **Sum**marization), a dataset consisting of 3,122 case judgments from multiple High Courts and the Supreme Court of India in English, along with the summaries in both English and Hindi, drafted by legal practitioners.

There have been previous efforts in compiling document-summary pairs in the legal domain, such as the *Eur-Lex* (Aumiller et al., 2022) dataset containing 1,500 document-summary pairs per language, for several European languages. However, to our knowledge, there has not been any similar effort in the Indian legal domain. Thus, MILDSum is the first dataset to enable cross-lingual summarization of Indian case judgments.

To construct MILDSum, we utilize the website *LiveLaw* (https://www.livelaw.in/), a popular site among Indian Law practitioners, that publishes articles in both English and Hindi summarizing the important case judgments pronounced by the Supreme Court and High Courts of India. According to the site, these articles are written by qualified Law practitioners; hence, they can be considered as high-quality summaries of the case judgments. We carefully extract the case judgments (in English) from this website, along with the English and Hindi articles summarizing each judgment. A major challenge that we had to address in this process is to link the English article and the Hindi article corresponding to the same judgment. The MILDSum dataset is available at https://github.com/Law-AI/MILDSum.

We also benchmark a wide variety of summarization models on the MILDSum dataset. We consider two broad approaches – (1) a *summarize-then-translate pipeline*, where a summarization model is used over a case judgment (in English) to generate

---

[*]Equal contribution by the first two authors

an English summary, and then the English summary is translated into Hindi, and (2) a *direct cross-lingual summarization* approach, where a state-of-the-art cross-lingual summarization model (Bhattacharjee et al., 2023) is finetuned over our dataset to directly generate Hindi summaries from the English documents. We observe that the summarize-then-translate approach performs better (e.g., best ROUGE-2 score of 32.27 for English summaries and 24.87 for Hindi summaries on average) than the cross-lingual summarization approach (best ROUGE-2 score of 21.76 for the Hindi summaries), in spite of reduction in summary quality due to the translation stage. Thus, our experiments demonstrate the need for better cross-lingual summarization models for the legal domain. To this end, we believe that the MILDSum dataset can be very useful in training and evaluating cross-lingual summarization models, for making legal judgments accessible to the common masses in India.

Also note that, apart from training summarizing models, the MILDSum dataset can also be used to train / evaluate Machine Translation (MT) models in the legal domain, given the paired English and Hindi summaries. MT in the Indian legal domain is another challenging problem that has not been explored much till date.

## 2 Related Works

There has been limited prior work on cross-lingual summarization in Indian languages. A popular large-scale multi-lingual summarization dataset *MassiveSumm* was introduced by Varab and Schluter (2021) which includes some Indian languages as well. The *ILSUM* dataset (Urlana et al., 2023) encompasses Hindi and Gujarati, in addition to English. Bhattacharjee et al. (2023) recently introduced a large-scale cross-lingual summarization dataset *CrossSum*, while Ladhak et al. (2020) released *WikiLingua* as a benchmark dataset for cross-lingual summarization. However, none of these datasets have specifically addressed cross-lingual summarization within the legal domain.

There exist some multi/cross-lingual summarization datasets in the legal domain for non-Indian languages, such as *Eur-Lex* (Aumiller et al., 2022) and *CLIDSUM* (Wang et al., 2022). In particular, *Eur-Lex* is a multi- and cross-lingual corpus of legal texts originating from the European Union (EU); it covers all 24 official languages of the EU. But to our knowledge, no such effort has been made in

| Domain | #Doc | Avg. #tokens | | |
|---|---|---|---|---|
| | | Doc | EN_Sum | HI_Sum |
| Court Judgments | 3,122 | 4,696 | 724 | 695 |

Table 1: Statistics of the MILDSum dataset

the Indian legal domain, and this work takes the first step to bridge this gap.

We refer the readers to Appendix A for more details about related work.

## 3 The MILDSum Dataset

In this work, we develop **MILDSum** (**M**ultilingual **I**ndian **L**egal **D**ocument **Sum**marization), a collection of 3,122 Indian court judgments in English along with their summaries in both English and Hindi, drafted by legal practitioners. Statistics of the dataset are given in Table 1. This section describes the data sources and methods applied to create MILDSum, as well as studies some important properties of the dataset.

### 3.1 Data Sources

The data for this study was primarily collected from *LiveLaw*, a reputed website known for its reliable coverage of court judgments in India. The site maintains one version in English[1] and another in Hindi[2]. Both versions of *LiveLaw* provide articles summarizing recent case judgments pronounced in the Indian Supreme Court and High Courts. According to the site, the articles (which are considered as the summaries) are written by qualified Law practitioners, which gives us confidence about the quality of the collected data.

### 3.2 Data Collection and Alignment

We observed that the articles in the English and Hindi websites of *LiveLaw* do *not* link to the corresponding articles in the other website. Hence the primary challenge is to pair the English and Hindi articles that are summaries of the same court judgment. We now describe how we address this challenge.

We observed that most of the Hindi article pages contain some parts of the titles of the corresponding English articles, if not the whole, in the form of metadata. We leveraged this information to find the matching English article for a particular Hindi article. To this end, we used the *Bing Search* API with a specialized query – *site: https://www.livelaw.in/ {English title metadata}* – to get the search results from *https://www.livelaw.in/* (the English site), and

---

[1] https://www.livelaw.in/
[2] https://hindi.livelaw.in/

then we considered the first result returned by the *Bing Search* API.

To ensure the correctness of the matching (between the Hindi and English articles), we computed the Jaccard similarity between the metadata title (obtained from the Hindi article) and the title on the first page fetched by the *Bing Search* API (the potentially matching English article). We then enforced these 3 conditions for accepting this match as a data-point – (i) If Jaccard similarity is greater than 0.8, then we accept the match, (ii) If Jaccard similarity is in between 0.5 and 0.8, then we go for manual checking (whether the two articles indeed summarize the same case judgment), and (iii) If Jaccard similarity is lower than 0.5, then we discard this data-point. Additionally, we require that at least one of the articles must contain a link to the original judgment (usually a PDF).

If the conditions stated above are satisfied, then we include the English judgment, the corresponding English article, and the corresponding Hindi article, as a data-point in MILDSum.

**Dataset cleaning:** The case judgments we obtained are in the form of PDF documents, while the English and Hindi articles are HTML pages. We employed various standard tools / Python libraries to extract the text from the downloaded HTML and PDF files. Particularly for the PDF documents, we utilized the *pdftotext* tool[3] that converts PDFs to plain text while preserving text alignment. More details about the text extraction process and pre-processing of the documents are given in Appendix B.1.

### 3.3 Quality of the dataset

As described earlier, the MILDSum dataset was constructed by matching English and Hindi articles summarizing the same judgment. Concerns may arise regarding the quality of this matching, as they were not cross-referenced on the websites. To address this concern, we took a random sample of 300 data-points and manually checked whether the English and Hindi summaries corresponded to the said judgment. Only one data-point was found erroneous, where the English and Hindi summaries corresponded to different judgments. Hence, we conclude that a very large majority ($> 99\%$) of the data-points in MILDSum are correct.

---

[3] https://github.com/jalan/pdftotext

### 3.4 Dataset statistics and analysis

MILDSum comprises a total of 3,122 case documents and their summaries in both English and Hindi. The average document length is around 4.7K tokens, the average English summary length is around 724 tokens, and the average Hindi summary length is around 695 tokens. The Compression Ratio in MILDSum – the ratio between the length of the English summaries to that of the full documents – is 1:6. The document length distributions of judgments and summaries are reported in Appendix B.2.

We observed that the *LiveLaw* articles (which are the reference summaries in MILDSum) are a mixture of extractive and abstractive summaries – they comprise of both verbatim quotations from the original case judgments (extractive parts) as well as paragraphs written in a simplified and condensed manner (abstractive parts). To quantify this extract/abstract-iveness of the summaries, we computed *Extractive Fragment Coverage* and *Extractive Fragment Density* as defined by Grusky et al. (2018). We get a high value of 0.90 for *Coverage*, which measures the percentage of words in the summary that are part of an extractive fragment from the document. We also observe a high value of 24.42 for *Density*, which quantifies how well the word sequence of the summary can be described as a series of extractions. These high values are expected as the summaries in our dataset frequently contain sentences directly quoted from the original judgments.

## 4 Experiments and Results

We divided the MILDSum dataset into 3 parts in a 70:15:15 ratio – *train set* (2185 data points), *validation set* (469 data points), and *test set* (468 data points). Only the *test* split is used to benchmark the performance of several state-of-the-art summarization models. We consider two broad approaches – (i) a summarize-then-translate pipeline approach, and (ii) a direct cross-lingual summarization approach – that are detailed next.

### 4.1 Summarize-then-Translate approach

In this pipeline approach, we first use a summarization model to generate an English summary from a given English case judgment, and then we translate the English summary to Hindi. In the first stage, we compare the performances of the following summarization models.

- **Unsupervised Extractive methods:** We use *LexRank* (Erkan and Radev, 2004) – a graph-based method that uses eigenvector centrality to score and summarize the document sentences; *LSA-based summarizer* (Yeh et al., 2005) that uses Singular Value Decomposition to project the singular matrix from a higher dimensional plane to a lower dimensional plane to rank the important sentences in the document; and *Luhn-summarizer* (Nenkova et al., 2011) which is a frequency-based method that uses TF-IDF vectors to rank the sentences in a document.[4]
- **Supervised Extractive models:** These models treat summarization as a binary classification task (whether to include a sentence in the summary), where sentence representations are learned using a hierarchical encoder. We use *SummaRuN-Ner*[5] (Nallapati et al., 2017) that uses two-layer bi-directional *GRU-RNN*. The first layer learns contextualized word representations which are then average-pooled to obtain sentence representations from the input document. Second, we use *BERTSumExt*[6] (Liu and Lapata, 2019) which takes pre-trained *BERT* (Devlin et al., 2019) as the sentence encoder and an additional Transformer as the document encoder.[7] Details of training these models on the training split of MILDSum are given in Appendix C.1.
- **Pretrained Abstractive models:** We use *Legal-Pegasus*[8], a fine-tuned version of *Pegasus* (Zhang et al., 2019) model, that is specifically designed for summarization in the legal domain by fine-tuning over the *sec-litigation-releases* dataset consisting of 2,700 US litigation releases & complaints. We also use two **Long Document Summarizers**. First, we use *LongT5* (Guo et al., 2022), an enhanced version of the *T5* (Raffel et al., 2019) model with Local and Transient Global attention mechanism than can handle long inputs up to 16,384 tokens. Next, we use *LED* (*Longformer Encoder Decoder*) (Beltagy et al., 2020), a Longformer variant with an attention

mechanism that scales linearly with sequence length, enabling the model to perform seq-to-seq tasks over long documents containing thousands of tokens.

**Translation of Generated summaries:** In the second stage of this pipeline approach, all the generated English summaries are translated into Hindi using the Machine Translation (MT) model *Indic-Trans* (Ramesh et al., 2022) which has been specifically trained for translation between English and Indian languages. We compared several MT models over a sample of our dataset, and observed *IndicTrans* to perform the best. More details on the translation experiments and the choice of *Indic-Trans* are given in Appendix C.2.

## 4.2 Direct Cross-Lingual Summarization

In this approach, we directly generate a Hindi summary for a given English document (without needing any translation). To this end, we use the state-of-the-art cross-lingual summarization model *CrossSum-mT5* (Bhattacharjee et al., 2023) which is a fine-tuned version of *mT5* (Xue et al., 2021). We specifically use the version that is fine-tuned over all cross-lingual pairs of the *CrossSum* dataset, where target summary is in Hindi.[9] In other words, this model is meant for summarizing text written in any language to Hindi.

## 4.3 Experimental setup

**Chunking of long documents:** Abstractive models like *Legal-Pegasus* and *CrossSum-mT5* have an input capacity of 1024 tokens and 512 tokens respectively; hence a case judgement often cannot be input fully into such a model. So, to deal with the lengthy legal documents, we chunked the documents into $m$ small chunks, where the size of each chunk is the maximum number of tokens (say, $n$) that the model is designed to accept without truncating (e.g., $n = 512$ for *CrossSum-mT5*). Then, we asked the model to provide a summary of $k$ tokens for each chunk, and append the chunk-wise summaries in the same order in which the chunks appear in the document, such that the combined summary (of total $m * k$ tokens) is almost equal in length to the reference summary.

Long document summarizers such as *LED* and *LongT5* have the input capacity of 16,384 tokens; hence, these models successfully handled almost

---

[4] We used the implementations of these unsupervised methods from https://github.com/miso-belica/sumy.

[5] https://github.com/hpzhao/SummaRuNNer

[6] https://github.com/nlpyang/PreSumm

[7] In the original *BERTSumExt*, there is a post-processing step called *Trigram Blocking* that excludes a candidate sentence if it has a significant amount of trigram overlap with the already generated summary to minimize redundancy in the summary. But, we observed that this step leads to too short summaries, as also observed by Sotudeh et al. (2020). Hence, we ignore this step in our work.

[8] https://huggingface.co/nsi319/legal-pegasus

[9] https://huggingface.co/csebuetnlp/mT5_m2o_hindi_crossSum

| Model | English Summary | | | Hindi Summary | | |
|---|---|---|---|---|---|---|
| | ROUGE-2 | ROUGE-L | BERTScore | ROUGE-2 | ROUGE-L | BERTScore |
| **Extractive – Unsupervised** | | | | | | |
| *LexRank* | 30.49 | 29.24 | 83.97 | 23.83 | 23.73 | 74.36 |
| *LSA-based* | 29.46 | 28.54 | 83.71 | 23.09 | 23.27 | 74.07 |
| *Luhn* | 28.76 | 28.12 | 83.58 | 22.46 | 22.93 | 73.80 |
| **Extractive – Supervised** | | | | | | |
| *SummaRuNNer* | **32.27** | **30.34** | 84.13 | **24.87** | **24.55** | 74.30 |
| *BERTSumExt* | 28.71 | 29.11 | 83.77 | 21.11 | 22.50 | 73.36 |
| **Abstractive** | | | | | | |
| *Legal-Pegasus-finetuned* | 31.84 | 28.36 | 84.14 | 24.40 | 22.58 | 74.62 |
| **Abstractive – Long Document Summarizer** | | | | | | |
| *LongT5-finetuned* | 23.25 | 23.47 | 83.90 | 21.32 | 20.34 | **75.70** |
| *LED-finetuned* | 25.51 | 27.81 | **84.20** | 19.56 | 22.51 | 74.73 |
| **Direct cross-lingual summarization (without translation)** | | | | | | |
| *CrossSum-mT5* | – | – | – | 8.75 | 15.86 | 70.55 |
| *CrossSum-mT5-finetuned* | – | – | – | 21.76 | 20.68 | 75.05 |

Table 2: Benchmarking different types of summarization methods over the MILDSum dataset. All scores are averaged over the *test* split of MILDSum. The best value of each metric is boldfaced.

all (∼96%) the case judgments in MILDSum without truncation or chunking. The few documents that longer were truncated at 16,384 tokens.

**Fine-tuning of abstractive models:** We finetune the abstractive models *Legal-Pegasus*, *LongT5*, *LED*, and *CrossSum-mT5* using the *train* and *validation* split of MILDSum. The method for generating finetuning data is explained in Appendix C.3.

**Hyperparameters:** For all models, we have used the default hyperparameters, since we wanted to benchmark their off-the-shelf performances over our dataset. We used a default seed value of 42 to initialize the model parameters before fine-tuning/training, to ensure that the same results can be reproduced with the same settings.[10]

**Evaluation metrics:** To evaluate the quality of the summaries generated by the models, we considered these standard metrics – *ROUGE* (Lin, 2004), and *BERTScore* (Zhang et al., 2020). Specifically, we reported the F1 scores of ROUGE-2, ROUGE-L, and BERTScore. More details on how the metrics are computed are reported in Appendix C.4.

### 4.4 Main Results

Table 2 reports the performance of all methods stated above, where all metric values are averaged over the *test* split of MILDSum. We see that the extractive method *SummaRunner* achieved the highest ROUGE scores, closely followed by *Legal-Pegasus-finetuned*. This is possibly because the reference summaries in MILDSum have a mix of extractive and abstractive features, and they frequently quote exact lines from the original judg-

ments (as stated earlier in Section 3.4). However, the abstractive long document summarizer methods (*LongT5-finetuned* and *LED-finetuned*) perform slightly better than extractive methods in terms of BERTScore.

The direct cross-lingual summarizer, *CrossSumm*, performed poorly when used off-the-shelf (without fine-tuning). But, *CrossSumm-finetuned* (fine-tuned over the *train* split of our MILDSum corpus) outperforms the off-the-shelf *CrossSumm* by a significant margin. This clearly shows the significance of our MILDSum corpus in direct cross-lingual summarization.

Overall, based on our experiments, the 'Summarize-then-Translate pipeline approach' performs better than the 'Direct Cross-Lingual Summarization' over MILDSum. Note that, in the pipeline approach, the translation stage usually introduces some additional errors, as shown by the consistently lower scores for Hindi summaries than the corresponding English summaries. In spite of the additional translation errors, the Summarize-then-Translate approach achieves higher scores over the MILDSum dataset. This result shows the need for improved cross-lingual summarization models for the legal domain in future.

### 5 Conclusion

This study develops the first multi- and cross-lingual summarization dataset for Indian languages in the legal domain (available at `https://github.com/Law-AI/MILDSum`). We also benchmark a variety of summarization models on our dataset. Our findings emphasize the need for better cross-lingual summarization models in the Indian legal domain.

---

[10]The experiments were carried out on a system having a NVIDIA V100 16GB GPU. The total GPU time needed for all fine-tuning experiments was approximately 48 hours.

## Limitations

One limitation of the MILDSum corpus is that it is developed entirely from one source (the *LiveLaw* site). However, sources that provide cross-lingual summaries in the Indian legal domain (i.e., English judgments and corresponding summaries in some Indian language) are very rare. Also note that our dataset covers a wide range of cases from diverse courts such as the Supreme Court of India, and High Courts in Delhi, Bombay, Calcutta, etc. Also, the different articles (summaries) were written by different Law practitioners according to the *Livelaw* website. Therefore, we believe that the dataset is quite diverse with respect to topics as well as in writers of the gold standard summaries.

Also, a good way of evaluating model-generated legal summaries is through domain experts. While an expert evaluation has not been carried out on the MILDSum dataset till date, we plan to conduct such evaluation of the summaries generated by different methods as future work.

Another limitation of the dataset is the lack of summaries in Indian languages other than Hindi. A practically important future work is to enhance the dataset with summaries in other Indian languages.

## Ethics Statement

As stated earlier, the dataset used in this work is constructed from the *LiveLaw* website, using content that is publicly available on the Web. According to the terms and conditions outlined on this website, content such as judgments, orders, laws, regulations, or articles can be copied and downloaded for personal and non-commercial use. We believe our use of this data adheres to these terms, as we are utilizing the content (mainly the judgments and the articles from which the summaries are obtained) for non-commercial research purposes. Note that the MILDSum dataset created in this work is meant to be used only for non-commercial purposes such as academic research.

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

# Appendix

## A Details of Related work

**Datasets for summarization in Indian languages:** Some recent work on cross-lingual summarization has been done by Bhattacharjee et al. (2023) who developed a large-scale cross-lingual summarization dataset *CrossSum* comprising 1.7M article-summary samples in more than 1.5K language-pairs. Another dataset *WikiLingua* was provided by Ladhak et al. (2020), which comprises of *Wiki-How* articles and their summaries in multiple languages, including a substantial collection of over 9K document-summary pairs in Hindi. However, note that the summaries in this dataset are relatively short, with an average length of only 39 tokens. This characteristic implies that the *Wik-iLingua* dataset is *not* ideal for training legal summarization systems, since legal summarization often necessitates longer summaries that encompass intricate legal concepts.

Notably, none of the datasets stated above are focused on cross-lingual summarization in the legal domain. This work is the first to develop a dataset for cross-lingual summarization of legal text in Indian languages.

**Datasets for cross-lingual summarization in non-Indian languages:** In the case of non-Indian languages, Scialom et al. (2020) proposed a dataset called *MLSUM*, which is compiled from online news sources. It comprises more than 1.5M pairs of articles and summaries in 5 languages: French, German, Spanish, Russian, and Turkish. Recently, Wang et al. (2022) released a benchmark dataset called *CLIDSUM* for Cross-Lingual Dialogue Summarization.

In the context of legal text summarization, Aumiller et al. (2022) recently introduced *Eur-Lex*, a multi- and cross-lingual corpus of legal texts and human-written summaries from the European Union (EU). This dataset covers all 24 official languages of the EU. To our knowledge, no such effort has been made in the Indian legal domain; we are taking the first step in bridging this gap.

## B Additional details about the MILDSum dataset

### B.1 Data pre-processing and cleaning

We used the *pdf2text* tool for extracting text from PDFs (the original case judgements). We conducted a quality check to ensure the reliability of the *pdf2text* tool. We randomly selected 5 pages from various documents and then manually compared the text present in the PDF with the text extracted using the *pdf2text* tool. We find that the tool's output closely matches the content present in the PDFs, with only minor instances of punctuation errors. Also, these judgment PDFs were of high quality, contributing to the tool's accurate performance.

We also observed the need for data cleaning / pre-processing while developing the MILDSum corpus. Case judgements usually contain some metadata or additional text at the beginning, which should *not* be part of the input for summarization. Figure 1 shows an example of the first page of a court case judgment, where the unnecessary text portions are highlighted in pink. Such text portions should be discarded and not be part of the input to a summarization model.

To clean this unnecessary part, we developed an algorithm that removes lines with fewer non-space characters than the average for a document, since the said lines mostly contain very few words. The algorithm proceeds as follows: compute each document's average number of non-space characters per line. Then, iterate through each line, deleting those lines with character counts below the average, unless the previous line was included. This ensures that the last line of each paragraph is retained. Using this algorithm, we discard the metadata and retain the content of the main order/judgment from the judgment PDFs.

### B.2 Document length distributions of judgments and summaries in MILDSum

This section describes the document length distributions (with respect to the number of words contained in the documents) of the case judgments and summaries in MILDSum. A document length distribution graph visually represents the number of documents in a dataset that contain a certain number of words. Each point on the graph represents a bin of document lengths (in terms of number of words) and the number of documents in that bin. The shape of the graph can vary depending on the dataset's characteristics.

Figure 2 shows the distribution of lengths of the case judgments in the MILDSum dataset. Similarly, Figure 3 and Figure 4 show the document length distributions of the English and Hindi summaries respectively. For our MILDSum corpus, the dis-

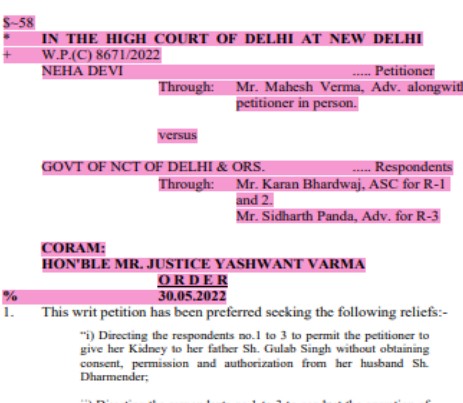

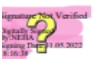

Figure 1: Example of the first page of a court judgment. The metadata (that is to be removed) is highlighted.

tribution for court judgements (Figure 2) seems to follow an *Inverse-Chi-Squared* distribution, while for both English and Hindi summaries, the distributions seem to follow *bell-shaped positively skewed* distribution curves.

## C More details about the experiments

### C.1 Training supervised extractive summarization models

The supervised extractive summarization models *SummaRuNNer* and *BERTSumExt* have been trained over the *training* split of the MILDSum dataset. These supervised methods require labeled data for training, where every sentence in the document must be labeled as 1 if this sentence is suitable for inclusion in the summary, and labeled as 0 otherwise. Hence, we convert the reference English summaries of MILDSum to purely extractive form to train these methods. To this end, we adopt the technique mentioned by Nallapati et al. (2017). Briefly, we assign label 1 (suitable for inclusion in the extractive summary) to those sentences from the full document that greedily maximize the ROUGE-2 (Lin, 2004) overlap with the human-written reference summary. The rest of the sentences in the full document are assigned label 0.

### C.2 Selection of translation model for translating English summaries to Hindi

For the second stage of the pipeline approach (Summarize-then-Translate), we tried out several state-of-the-art Machine Translation (MT) systems for English-to-Hindi translation, such as *Google Cloud Translator*[11], *Microsoft Azure Translator*[12], *IndicTrans*[13] (Ramesh et al., 2022), *mBART-50*[14] (Tang et al., 2020), *NLLB*[15] (team et al., 2022), and *OPUS*[16] (Tiedemann and Thottingal, 2020). To compare among these MT models, we randomly selected 100 data-points from MILDSum, and then used these MT models for English-to-Hindi translation of the reference English summaries. We used the standard MT evaluation metric *BLEU* (Papineni et al., 2002) for comparing the quality of the translations, by matching a machine-translated Hindi summary with the reference Hindi summary for the same case judgment.

Among these MT systems, **IndicTrans** exhibited the best performance with a *BLEU* score of 50.34 (averaged over the 100 data-points randomly selected for this evaluation). The scores for other models are as follows – *MicrosoftTrans*: 47.10, *GoogleTrans*: 43.26, *NLLB*: 42.56, *mBART-50*: 33.60, and *OPUS*: 10.65. Also, *IndicTrans* was found to be the best for English-to-Hindi translation in the prior work (Ramesh et al., 2022). For these reasons, we used *IndicTrans* for translation of the English summaries to Hindi.

### C.3 Generating fine-tuning data for abstractive summarization models

This section describes how we created fine-tuning data for the abstractive models *Legal-Pegasus* and *CrossSum-mT5*. Fine-tuning of these models requires data in the form of document chunks and the corresponding summaries of the chunks. For this experiment, we fixed the chunk length to 512 tokens. As we are chunking a case judgment, we need to chunk the corresponding reference summary as well to match the context of the judgment's

---

[11]https://cloud.google.com/translate/docs/samples/translate-v3-translate-text
[12]https://azure.microsoft.com/en-us/products/cognitive-services/translator
[13]https://github.com/AI4Bharat/indicTrans
[14]https://huggingface.co/facebook/mbart-large-50-one-to-many-mmt
[15]https://huggingface.co/facebook/nllb-200-3.3B
[16]https://huggingface.co/Helsinki-NLP/opus-mt-en-mul

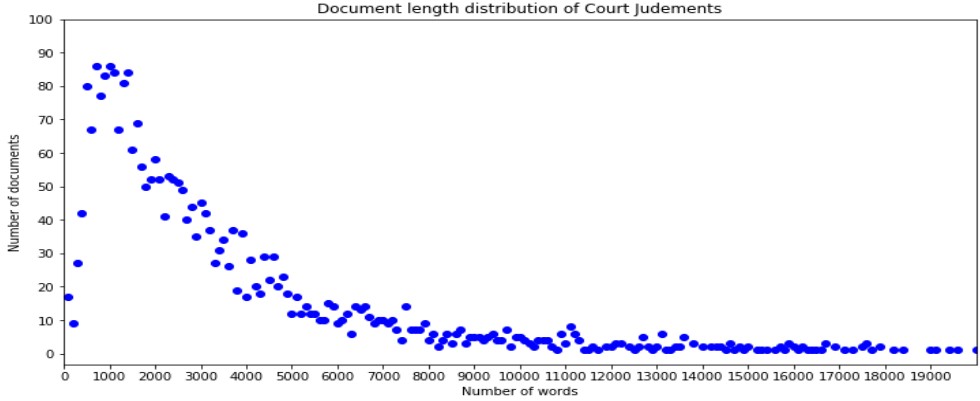

Figure 2: Document length distribution of court judgments in MILDSum

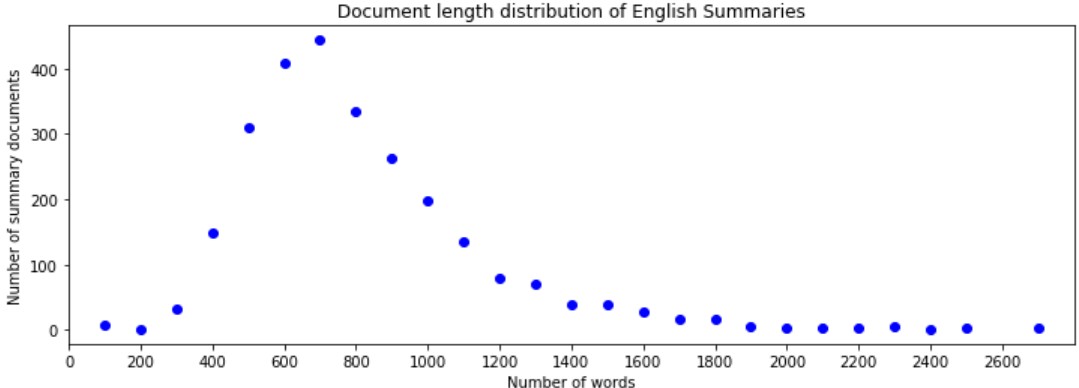

Figure 3: Document length distribution of English summaries in MILDSum

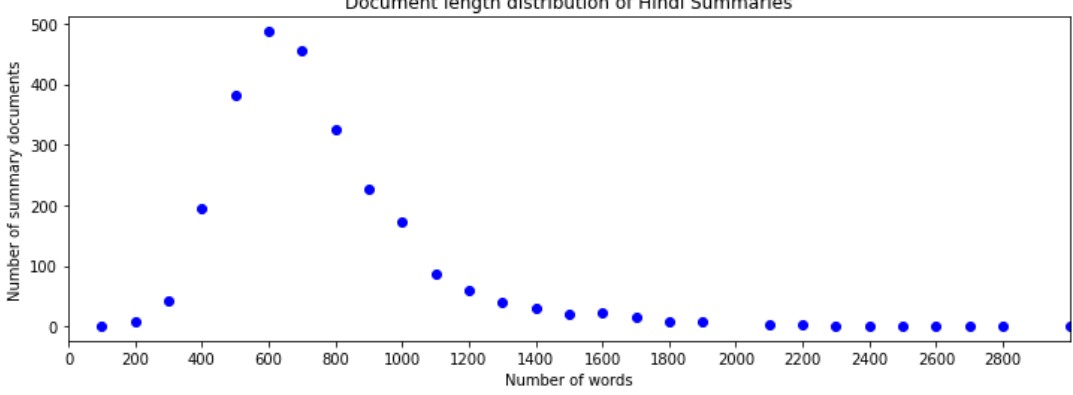

Figure 4: Document length distribution of Hindi Summaries in MILDSum

chunks. In order to generate parallel pairs (document chunk and corresponding summary chunk), we followed the method of Reimers and Gurevych (2020) i.e., *bitext mining*. We generate embeddings using *LaBSE* (Feng et al., 2022). Then, we use the following scoring function to compute a score for each pair of embeddings; we set the threshold to 1 for including a given pair in the fine-tuning dataset.

$$score(x, y) = margin\Big\{cos(x, y),$$

$$\sum_{z \epsilon NN_k(x)} \frac{cos(x, z)}{2k} + \sum_{z \epsilon NN_k(y)} \frac{cos(y, z)}{2k}\Big\}$$

Where $x$ and $y$ are the embeddings of the judgment chunk and summary chunk respectively. $NN_k(x)$ denotes the $k$ nearest neighbors of $x$. We have used ratio-based margin as the margin function, i.e., $margin(a, b) = a/b$.

### C.4 Evaluation metrics for summarization performance

The following standard metrics were used to evaluate the quality of the model-generated summaries, by comparing with the reference summaries.

- **ROUGE** (Lin, 2004) stands for *Recall-Oriented Understudy for Gisting Evaluation*. ROUGE-2 measures the textual overlap (bi-grams) between the model-generated summaries and the reference summaries. ROUGE-L measures the longest matching sequence of words using the Longest Common Subsequence (LCS).

  To calculate multilingual ROUGE scores (English and Hindi in this work), we used the *multilingual_rouge_scoring* library[17] (provided by Hasan et al. (2021)) which uses the OpenNMT tokenizer[18].

- **BERTScore** (Zhang et al., 2020) uses *BERT* to compute the similarity scores between the token-level embeddings of the model-generated and reference summaries. It is known to correlate better with human judgments as it considers the semantic part.

  To calculate multilingual BERTScore, we used the *Huggingface evaluate* library[19] that incorporates the official BERTScore project[20].

---

[17]https://github.com/csebuetnlp/xl-sum
[18]https://opennmt.net/
[19]https://huggingface.co/spaces/evaluate-metric/bertscore
[20]https://github.com/Tiiiger/bert_score