# OpenReview forum: "MILDSum: A Novel Benchmark Dataset for Multilingual Summarization of Indian Legal Case Judgments"
_EMNLP/2023/Conference — EMNLP 2023 Main_

### Official Review · Reviewer_eZYq · 2023-08-01

**Soundness:** 4

**Excitement:**

3: Ambivalent: It has merits (e.g., it reports state-of-the-art results, the idea is nice), but there are key weaknesses (e.g., it describes incremental work), and it can significantly benefit from another round of revision. However, I won't object to accepting it if my co-reviewers champion it.

**Paper Topic And Main Contributions:**

The authors address in their work an important problem - making legal case judgments given by Indian courts accessible to the Indian populace. While the case studies are usually written in complex English, the English proficiency of many of the country's people is not high enough to be able to read and understand them. Therefore, the authors wish to develop automatic methods to summarise the English cases in Hindi.

For this reason, the authors collect a cross-lingual dataset of legal cases and their summaries, and benchmark it. They find that first summarising and then translating is the best approach in practice.

**Questions For The Authors:**

1) What is the datasets' license?
2) Did you validate the quality and error rate of the pdf2text tool?
3) According to line 294, it seems like you didn't finetune CrossSum-mT5 on your dataset. Is this correct? If so, why?
4) Did you fine tune several models with several seeds? what is the confidence score of your evaluation?


**Reasons To Accept:**

1) The authors tackle a real-world important problem, that doesn't have existing good solutions. The authors collected a decently sized dataset and will release it.
2) The authors manually evaluated the quality of a large partition of the collected dataset.
3) The authors benchmark the dataset using popular and reasonable models.


**Reasons To Reject:**

1) Related work section is too short. But I expect this issue can be easily fixed given the extra page of the camera ready, as more relevant info is provided in the appendix.
2) There are many presentation and writing issues. The paper must be better proofread.
3) While the authors tackle a real-life problem that affects many people, it is still quite limited, and not many researchers will benefit from the release of the dataset.

**Reproducibility:**

3: Could reproduce the results with some difficulty. The settings of parameters are underspecified or subjectively determined; the training/evaluation data are not widely available.

**Reviewer Confidence:**

4: Quite sure. I tried to check the important points carefully. It's unlikely, though conceivable, that I missed something that should affect my ratings.

**Typos Grammar Style And Presentation Improvements:**

- line 24) write what the interesting inferences that you found are, otherwise the abstract is incomplete
- line 132) what date mismatch? In general, this entire subsection is confusing and unclear
- The paper clearly requires more proofreading, there are many mistakes throughout
- The introduction contains too many superlatives in my opinion
- The underlined and bold text in section 4.1 is a bit much

---

> ### Author Rebuttal · Authors · 2023-08-29
>
> Thank you for your time and effort in giving us valuable feedback and for recognizing our efforts in this work. We refer to your “reasons to reject” as “RR”, and your questions as “Q” below.
>
> RR1 (Related Work): Yes, we could not give details in the main paper due to lack of space. We will expand the section in the final version.
>
> RR2 (Writing issues): We apologize for the issues in writing. We will proofread the paper thoroughly and correct the issues.
>
> RR3: While we acknowledge your perspective, we would like to clarify a few points. It is important to highlight the specific target audience and the broader impact of our work. This study is intended for researchers who are interested in Summarization and Legal-NLP. In a linguistically diverse country like India, where legal texts are written in complex English due to historical reasons, it is crucial to summarize them in Indian languages to ensure equitable access to justice. So, recognizing India's linguistic diversity and the need for improving access to justice, multilingual summarization datasets are very much required in the Indian legal domain. This study presents a pioneering effort towards cross-lingual summarization of English legal documents into Hindi, the most frequently spoken Indian language. Also, we are committed to enriching MILDSum's impact by including summaries in other Indian languages.
>
> Additionally, please note that MILDSum is large enough compared to other cross-lingual legal datasets (non-Indian) like Eur-Lex [https://aclanthology.org/2022.emnlp-main.519/]. Eur-Lex contains 1,500 doc-summ pairs per language. Although language diversity is less in MILDSum compared to Eur-Lex, MILDSum contains 3,122 cross-lingually aligned doc-summ pairs that is much higher than Eur-Lex. So, we believe MILDSum is a valuable resource for researchers in this domain. We also envision to evaluate the performances of more recent Large Language Models (LLMs) such as LLaMA and GPT-4 on our proposed legal domain-specific dataset in the future.
>
> Also note that, apart from training summarizing models, this dataset can also be used to train / evaluate Machine Translation (MT) models, given the paired English and Hindi summaries. MT in the Indian legal domain is another challenging problem which has not been explored much. We will point out this additional utility of the dataset in the final version.
>
> [Q1] The dataset used in this work is primarily constructed from the "LiveLaw" website. According to the terms and conditions outlined on the website, content such as judgments, orders, laws, regulations, or articles can be copied and downloaded for personal and non-commercial use. Our use of this data adheres to these terms, as we are utilizing the content (mainly the judgements and the articles from which the summaries are obtained) for non-commercial research purposes.
>
> [Q2] We have now conducted a quality check to ensure the reliability of the pdf2text tool. We randomly selected 5 pages from various documents and then manually compared the text present in the PDF with the text that was extracted using the pdf2text tool. We find that the tool's output closely matches the content present in the PDFs, with only minor instances of punctuation errors. Also, these judgment-PDFs were of high quality, contributing to the tool's accurate performance. We will add this quality check in the camera-ready draft.
>
> [Q3] We acknowledge that we did not report the results of fine-tuned CrossSum on our dataset. We attempted to fine-tune the CrossSum-mT5 model (source code and English-to-Hindi checkpoint obtained from https://github.com/csebuetnlp/CrossSum/ and https://huggingface.co/csebuetnlp/mT5_m2o_hindi_crossSum). We obtained fairly low scores even after fine-tuning the model for a few epochs (using default hyperparameters). Upon careful observation, we found that the fine-tuned model was frequently hallucinating facts and producing repetitive summaries. Hence, we avoided reporting the scores in the paper. The scores we got:
> [Scores of off-the-shelf version (reported in paper) - Rouge-2: 6.54, Rouge-L: 14.45, BERTScore: 69.49]
> [Scores of fine-tuned version - Rouge-2: 4.15, Rouge-L: 14.85, BERTScore: 62.81]
> We will further investigate the issue, and try to add the results of fine-tuned CrossSum-mT5 in the final version.
>
> [Q4] We used a default seed value of 42 to initialize the model parameters before fine-tuning/training begins, to ensure that the same results can be reproduced every time with the same settings. We mentioned in L248 of the paper that we have used the default hyperparameters for all models. As suggested, we will run the models with different seed values and report the average values with confidence scores in the final version.
>
> Also, thank you for highlighting the presentation and writing issues. We will definitely rectify these issues in the final version.

---

### Official Review · Reviewer_XjyQ · 2023-08-02

**Typos Grammar Style And Presentation Improvements:** abstract/L024
**Soundness:** 3

**Excitement:**

3: Ambivalent: It has merits (e.g., it reports state-of-the-art results, the idea is nice), but there are key weaknesses (e.g., it describes incremental work), and it can significantly benefit from another round of revision. However, I won't object to accepting it if my co-reviewers champion it.

**Missing References:**

In the related work section it may be worth mentioning the few existing summarization datasets that do contain Indian languages:
- ILSUM (Urlana et al, 2023 - https://arxiv.org/abs/2303.14461)
- TeSum (Urlana et al, 2022 - https://aclanthology.org/2022.lrec-1.614)
- MassiveSumm (Varab and Schluter, 2021 - https://aclanthology.org/2021.emnlp-main.797/)

**Paper Topic And Main Contributions:**

This paper introduces the dataset MILDSum (Multilingual Indian Legal Document Summarization), a summarization dataset comprised of 3122 Indian legal case judgments. The dataset is comprised of aligned document-summary pairs in both English and Hindi (thus a total of 6.2k document-summary pairs). Documents and summaries are collected from the website LiveLaw, a reputable site for coverage of court judgments in India.

The paper includes a description of the data collection methodology which includes two pipelines to collect data and approaches to align and clean documents and summaries. 300 documents were sampled and manually checked to ensure cross-lingual alignment - only 1 document was found to be misaligned. The paper includes baseline results using 9 models, including both unsupervised, supervised, extractive, abstractive, and cross-lingual systems. The monolingual abstractive systems are fine-tuned on a training split of the proposed dataset, the cross-lingual system is an off-the-shelf system trained on newswire data (CrossSumm), and details about the monolingual extractive systems are not included. Baseline outputs on a held-out test set are evaluated using ROUGE and BERTScore. All models give similar scores except the cross-lingual model which performs very poorly.

**Questions For The Authors:**

- title: The title describes the dataset as multilingual, however, isn't it a cross-lingual dataset since the languages are aligned?
- L039: "our dataset includes detailed quotations from the ..." What does this mean?
- L069: "the heterogenous nature nature of our dataset" What does this mean?
- L235: SummaRuNNer is a little bit dated? Why not use something more performant like BERSUMExt? (https://aclanthology.org/D19-1387/)
- L278: The paper mentions that IndicTrans is best according to BLEU. What does this refer to? Did you compute BLEU yourself?
- L281: "Indic is known to be the best English-to-Hindi translation". Such a claim lacks a citation or supporting evidence.
- L299: ROUGE and BERTScore are parametric measures. What tokens are ROUGE computed over? What model is BERTScore computed. over? The standard is English, but your results are on Hindi.
-

**Reasons To Accept:**

Summarization datasets with high-quality reference summaries are highly desirable as existing datasets are often automatically collected from online sources where it is not always clear whether reference summaries are in fact satisfying summaries. This paper presents a dataset that builds on a reputable source where summaries appear to be of high quality, in addition to being distributed in two languages, and aligned across these languages. One of the languages, Hindi, has poor coverage in existing summarization resources and is the first dataset to cover the legal Hindi domain, providing a novel and valuable resource for Hindi and legal NLP.

**Reasons To Reject:**

The dataset is valuable and will undoubtedly be a much-welcomed resource that will benefit the community, however, I do find that the paper, in its current state, lacks some rigor and depth. In particular, very little information about the documents and summaries in the dataset is described. Outside the lengths of the average length of documents and summaries and brief mentions of the dataset being "extractive" due to quotations, it is unclear what the dataset looks like. I think the paper would benefit from some quantitative measures to help the reader understand the shape of the dataset. E.g. extract/abstract-iveness using measures like coverage/density/compression (grusky et al, 2018), what is the variation in document/summary length, vocabulary size, etc. Similarly, the included baselines experiment is confusing and light on the details. Some models are trained on a training split of the dataset, while others are not. Details about how the best model (SummaRuNNer) is not described. The evaluation measures ROUGE and BERTScore can be used in other languages but it is crucial to specify how they are adapted to Hindi (e.g. what tokens are ROUGE scores computed over? and which model is used to compute BERTScores?). The baseline results are very similar across all models, thus not telling the reader very much about what type of model is suitable for the data. Choosing fewer, more motivated models might improve the message of the experiment.

**Reproducibility:**

3: Could reproduce the results with some difficulty. The settings of parameters are underspecified or subjectively determined; the training/evaluation data are not widely available.

**Reviewer Confidence:**

3: Pretty sure, but there's a chance I missed something. Although I have a good feel for this area in general, I did not carefully check the paper's details, e.g., the math, experimental design, or novelty.

---

> ### Author Rebuttal · Authors · 2023-08-29
>
> Thank you for giving us valuable feedback and for appreciating the importance of our dataset. We denote your comments by “C” and our answers with “A”.
>
> C: “very little information about the documents and summaries…”
>
> A: Please note that several statistics about the dataset, that you suggested to include, are already reported in Appendix A.2. Please refer to Figures 2, 3 & 4 of the paper. Figure 2 shows the distribution of document lengths, Fig 3 and Fig 4 show the distributions of length of English and Hindi summaries respectively.
>
> C: “paper would benefit from some quantitative measures … E.g. extract/abstract-iveness”
>
> A: Following your suggestions, we computed the coverage, density, and Compression ratio to measure extract/abstract-iveness of the dataset. We find Extractive Fragment Coverage as 0.90, Extractive Fragment Density as 24.42, and Compression ratio as 5.84 for our dataset. The high values of Coverage and especially Density indicate high extractiveness of the summaries with respect to the source documents. This is expected, since summaries in our dataset contain several quoted lines from the original judgments (as already stated in lines 310-313). These values also justify why extractive summarization models tend to perform better than abstractive summarization models on this dataset. We will include these details in the final version.
>
> C: “The evaluation measures ROUGE and BERTScore … how they are adapted to Hindi”
>
> A: For multilingual ROUGE scores (in our case English and Hindi), we have used this library https://github.com/csebuetnlp/xl-sum [provided by https://aclanthology.org/2021.findings-acl.413/] which uses the OpenNMT tokenizer (https://opennmt.net/). To calculate multilingual BERTScore, we have used the 'evaluate' library (https://huggingface.co/spaces/evaluate-metric/bertscore) that incorporates the official BERTScore project (https://github.com/Tiiiger/bert_score). We will add these details in the final version.
>
> C: “Details about how the best model (SummaRuNNer) is not described.”
>
> A: We trained the vanilla SummaRuNNer model, obtained from https://github.com/hpzhao/SummaRuNNer, on our training data. We will include more details about the summarization models used, in the final version.
>
> C: “Choosing fewer, more motivated models might improve the message of the experiment.”
>
> A: We applied a few representative summarization models from different families - unsupervised extractive, supervised extractive (where we will add BERTSumExt too), abstractive (pre-trained and fine-tuned), and abstractive long document summarizers. Most of the supervised models have been trained / fine-tuned on the MILDSumm training set. The experiments show extractive models to perform slightly better than abstractive models, since the gold standard summaries often contain quotes / sentences extracted from the source documents. Another take-away is that the summarize-then-translate pipeline performs better than the cross-lingual summarization model. We will explain these takeaways more clearly in the final version.
>
> C: Query about paper title
>
> A: MILDSum contains 3,122 Indian court judgments in English along with their summaries in both English and Hindi. Since the dataset contains summaries in two languages, we considered this dataset as Multilingual.
>
> C: “L039: our dataset includes detailed quotations ..”
>
> A: We meant to say that MILDSum contains the actual judgments in their original form as provided by the respective courts. We will rephrase the sentence and clarify what we meant.
>
> C: “L069: the heterogeneous nature of our dataset"
>
> A: By the “heterogeneous nature of our dataset”, we meant that our gold standard summaries contain a mixture of directly quoted lines from the original judgments (extractive parts) as well as abstractive parts. Hence, the summaries are a heterogeneous mixture of extractive and abstractive parts. We will rephrase the sentence and clarify what we meant.
>
> C: “L235: Why not use something more performant like BERSUMExt?”
>
> A: Thank you for this suggestion. We assessed the performance of BERTSumExt as well. We trained it using the default settings on the training split of MILDSum and then tested it on the test split of MILDSum. The scores for the automatic metrics are as follows (F1 scores):
> Scores for English Summary - ROUGE-2: 32.08 , ROUGE-L: 31.12 and BERTScore: 86.78
> Scores for Hindi Summary - ROUGE-2: 24.34 , ROUGE-L: 25.62 and BERTScore: 74.59
>
>
> C: L278, L281: Queries about IndicTrans
>
> A: Yes, IndicTrans was observed to be the best translator from English to Hindi according to our internal experiments (which are briefly described in Appendix C). We tried out several Machine Translation (MT) systems such as Google-Translator, Microsoft-Translator, IndicTrans, mBART, NLLB, and OPUS over randomly selected 100 data points from MILDSum. We used the standard MT evaluation metric BLEU for comparing the quality of translations. Among these MT systems, IndicTrans exhibited the highest performance with a BLEU score of 50.34 [scores for other models are as follows -- MicrosoftTrans: 47.10, GoogleTrans: 43.26, NLLB: 42.56]. Also, IndicTrans was found to be the best MT system for English-to-Hindi translation in the prior work https://aclanthology.org/2022.tacl-1.9/. We mentioned briefly that we tried out these MT systems in Appendix C. We will add the BLEU scores and these details in the final version of the paper.
>
> Thank you for suggesting the additional references and typos. We will definitely incorporate these in the final version. Overall, we are happy to address your concerns/suggestions. Also, we would like to answer any follow-up questions that you may have.

---

### Official Review · Reviewer_egHn · 2023-08-04

**Typos Grammar Style And Presentation Improvements:** 1. Sometimes you use "English & Hindi…
**Soundness:** 4

**Excitement:**

4: Strong: This paper deepens the understanding of some phenomenon or lowers the barriers to an existing research direction.

**Paper Topic And Main Contributions:**

The paper presents a study on the automatic summarization of legal case judgments in India, particularly focusing on translating these summaries from English, the language in which most judgments are written, to Hindi to benefit a broader audience. The core contributions of the paper are:

1. The introduction of MILDSum (Multilingual Indian Legal Document Summarization), a first-of-its-kind dataset that consists of 3,122 judgment-summary pairs in both English and Hindi. This data is derived from several high-profile courts, including the Supreme Court of India. The summaries have been meticulously curated by legal practitioners, ensuring both their linguistic quality and legal accuracy.

2. Unlike previous works that predominantly focused on summarizing in the source language, this paper emphasizes the importance of cross-lingual summarization. This responds to the linguistic reality of India, where many citizens do not have proficient English skills but need access to legal judgments for various reasons.

3. The paper sets its dataset apart from existing European datasets like Eur-Lex, emphasizing the unique characteristics of Indian legal documents, including length, specialized terminologies, and referencing patterns.

4. The authors benchmarked various summarization models on the MILDSum dataset. Interestingly, they found that extractive methods outperformed abstractive ones, a finding they attribute to the specific nature of their dataset.

In conclusion, the paper addresses an important gap in the field of automatic summarization, especially in the context of the Indian legal system. By offering a new dataset and shedding light on the unique requirements of cross-lingual summarization in this domain, the study makes a significant contribution that holds potential for both academic and practical applications.

**Questions For The Authors:**

A. I am concerned about the data collection and alignment processes described in Sections 3.1 and 3.2. You mentioned that LiveLaw's Hindi site is "not frequently updated as its English counterpart" and the sites do not link articles in the two languages. How did you ensure that this lack of direct linking did not bias your data set toward the judgments published in English?

B. The "Summarize-then-Translate pipeline approach" and the "Direct Cross-Lingual Summarization" method were both introduced in the paper. Based on your experiments, can you clarify each method's specific advantages and disadvantages?

C. Your abstract states that you "benchmark the performance of several competitive summarization approaches on our corpus." However, the provided content does not go into detail about these benchmarks. What specific metrics did you use to evaluate the quality and accuracy of the summaries? Were there any human evaluators involved in this process?

D. I noticed that you've trained various models on the MILDSum dataset. How does the MILDSum dataset compare to other similar datasets (like Eur-Lex) in terms of challenges and opportunities for model training?

E. In Section 3.4, you mention that the dataset error rate is less than one percent. While this is commendable, can you shed light on the potential impact of this error on the validity of the research? Would there be any specific biases introduced by this error rate?

F. In terms of future applications: How can the MILDSum dataset aid in improving access to legal information for the general public in India? What are the potential implications of this research in broader legal tech domains?

G. The paper states that "extractive methods showcased superior performance compared to existing abstractive methods." Can you provide some insights into why extractive methods might be more suited for legal document summarization, especially in the Indian context?

H. You've mentioned the use of Bing search API for data collection. Were there any considerations or challenges you faced in ensuring data privacy and security during this data collection process?

I. Finally, while you have done an extensive job in curating this dataset, how do you plan on updating and maintaining MILDSum in the future to ensure its relevance and utility?

**Reasons To Accept:**

1. The introduction of the MILDSum dataset is a groundbreaking effort, given that no such dataset previously existed for multilingual summarization in the Indian legal domain. The meticulous curation of 3,122 judgment-summary pairs in both English and Hindi by legal practitioners ensures a high-quality resource for future research.

2. With a significant portion of India's population lacking proficient English skills but needing access to legal judgments, the paper addresses a pressing real-world problem. This has immense societal and practical implications, especially in a multilingual country like India.

3. The paper doesn’t just introduce a dataset; it delves deep into its characteristics, highlighting the unique features of Indian legal documents. This offers insights into domain-specific challenges and opportunities.

4. By testing various summarization models, the paper sets a benchmark for future research. The finding that extractive methods outperformed abstractive ones on this dataset provides a crucial insight for model selection and development in this domain.

Benefits to the NLP Community:

1. Given the dearth of resources and research on Indian legal summarization, this paper offers a fresh avenue for NLP researchers to explore, innovate, and develop applications.

2. Emphasizing the importance of cross-lingual summarization, this paper can act as a catalyst for similar initiatives in other multilingual countries, fostering global research collaboration.

3. With the rise of legal tech solutions globally, the insights and dataset from this paper can propel the development of tools that make legal judgments more accessible to a broader audience.

4. The methodology and considerations presented in this paper can serve as a blueprint for researchers looking to develop similar datasets in other specialized domains, further enriching the NLP research landscape.

**Reasons To Reject:**

Weaknesses:

1. The dataset, MILDSum, is primarily constructed from the website "LiveLaw". Depending entirely on one source may limit the diversity and generalizability of the dataset.

2. The described pipelines, especially when matching articles from Hindi and English versions of LiveLaw, are intricate and may be prone to errors.

3. While the paper mentions translating English summaries into Hindi, the quality of these translations isn't explicitly discussed. Given the intricacies of legal language, ensuring high-quality translations is paramount.

4. The paper benchmarks performance using MILDSum, but a comparative analysis using other datasets would make the results more robust. This could help establish the advantages of MILDSum over existing resources.

5. While Hindi is one of the most spoken languages in India, India is home to several other languages. A true multilingual dataset would encompass multiple Indian languages, broadening its applicability.

6. While 3,122 judgment-summary pairs can be considered a reasonable size, it might still be limited when training sophisticated deep learning models.

7. The models have been trained and tested only on MILDSum. Their performance on external datasets or real-world applications isn't discussed.

8. The approach to handle long legal documents by chunking might lead to the loss of context or crucial information.

Risks for Conference Presentation:

1. Even a less than 1% error rate in a legal dataset can have serious implications. Any inaccuracies in the legal domain can be consequential, and attendees might raise concerns.

2. Given the complexities in data collection and alignment, presenters should be well-prepared to address detailed technical questions regarding the methodology.

3. Audience members familiar with the Indian legal system or multilingual summarization might question the dataset's applicability across various domains or other Indian languages.

4. Any data collection activity, especially from a legal domain, should uphold ethical considerations. If these aren't explicitly mentioned in the paper, the presenters might face related queries.

5. Attendees might want to know how this research can be used practically, especially in real-world legal settings in India.

**Reproducibility:**

4: Could mostly reproduce the results, but there may be some variation because of sample variance or minor variations in their interpretation of the protocol or method.

**Reviewer Confidence:**

4: Quite sure. I tried to check the important points carefully. It's unlikely, though conceivable, that I missed something that should affect my ratings.

---

> ### Author Rebuttal · Authors · 2023-08-29
>
> Thank you for giving us valuable feedback and for appreciating our work. You have pointed out 8 Weaknesses (1-8) and gave 9 questions (A-I). We are trying to address them below. To keep the rebuttal brief, we have combined some related weaknesses and questions.
>
> [1] We understand the concern about using just one source ("LiveLaw") for our dataset, MILDSum. However, sources that provide cross-lingual summaries in the legal domain (English judgments and corresponding Hindi summaries) are very rare in India, and we did not find any other website of this type. Also please note that our dataset covers a wide range of cases from diverse courts such as the Supreme Court of India, and High Courts in Delhi, Bombay, Calcutta, etc. Also, different summaries were written by different legal experts. Therefore, we believe that the dataset is diverse with respect to topics as well as in writers of the gold standard summaries.
>
> [2, A, E] We agree that the pipeline for matching articles from Hindi and English versions of LiveLaw are intricate. This is one of the biggest challenges in developing the dataset, and we designed the pipeline very carefully (as described in Section 3.2 and 3.3 in the paper). As reported in Section 3.4, to check the correctness of pairing Hindi and English summaries, we conducted a careful quality check on 300 random data points of MILDSum and found just one mismatched instance. So we believe our matching method is very accurate. Also, given that we found only one error, it is difficult to comment on any particular biases introduced by this very small error rate.
>
> [3] Regarding English-to-Hindi translation, we tried out several Machine Translation (MT) systems such as Google-Translator, Microsoft-Translator, IndicTrans, mBART, NLLB, and OPUS over randomly selected 100 data points from MILDSum. We used the standard MT evaluation metric BLEU for comparing the quality of translations. Among these MT systems, IndicTrans exhibited the highest performance with a BLEU score of 50.34 [scores for other models are as follows -- MicrosoftTrans: 47.10, GoogleTrans: 43.26, NLLB: 42.56]. Also, IndicTrans was found to be the best MT system for English-to-Hindi translation in the prior work https://aclanthology.org/2022.tacl-1.9/. We mentioned briefly that we tried out these MT systems in Appendix C. We will add the BLEU scores and these details in the final version of the paper.
>
> [4, 7] Since MILDSum is the first cross-lingual summarization dataset for legal domain in Indian languages, we could not compare model performances over MILDSum with that over any other dataset in the legal domain. But we acknowledge the importance of comparative analysis using other non-legal datasets such as CrossSum (https://aclanthology.org/2023.acl-long.143/). We will include this comparative analysis in the final version.
>
> [5, 6, D, I] We fully acknowledge the value of a multilingual summarization dataset in the Indian legal domain. But please note that legal domain-specific data sources with cross-lingual nature are very rare. We found only one source - “LiveLaw” - from where we were able to obtain 3,122 judgment-summary pairs till date. Note that, with 3,122 judgment-summary pairs, MILDSum is large enough for training and fine-tuning summarization systems to improve their performances. Additionally, MILDSum is large enough compared to other cross-lingual legal datasets (non-Indian) like Eur-Lex (https://aclanthology.org/2022.emnlp-main.519/). Eur-Lex contains 1,500 doc-summ pairs per language. Although language diversity is less in MILDSum compared to Eur-Lex, MILDSum contains 3,122 cross-lingually aligned doc-summ pairs that is much higher than Eur-Lex. Similar to Eur-Lex, one of the most obvious challenges for this dataset is the extreme document lengths, and also length disparity between documents. Finally, we are committed to enriching MILDSum's impact by including summaries in other Indian languages in the future, as well as increasing the number of document-summary pairs by periodic crawling of the Livelaw websites.
>
> [8] We agree that chunking might lead to the loss of context. But, please note we have also used LongT5 and LegalLED, which can handle most of our documents in full; so chunking was not required for these models. We already stated this in lines 263 to 269 of the paper.
>
> [B] Based on our experiments, "Summarize-then-Translate pipeline approach" performs better than "Direct Cross-Lingual Summarization" over MILDSum. However the pipeline approach has the problem that errors in the summarization stage (e.g., omission of important information) are propagated to the second stage (translated). On the other hand, "Direct Cross-Lingual Summarization" models are free from such risks of error propagation; also these methods can be improved in the future.
>
> [C] For automatic evaluation, we have used the standard metrics ROUGE-2, ROUGE-L and BERTscores (as reported in Section 4.3). "Expert Evaluation" could not be done in this work due to unavailability of Law experts with knowledge of Hindi. We plan to conduct an  "Expert Evaluation" of the summaries generated by different methods as a future work.
>
> [F] As stated in the Introduction of the paper, most legal text in the Indian judiciary is written in complex English. But, a significant portion of India's population lacks a strong command in English. Hence, it is crucial to make legal knowledge available in Indian languages, of which Hindi is the most commonly spoken/understood language. While prior research primarily focuses on summarizing legal case judgments in their source languages, this study presents a pioneering effort toward cross-lingual summarization of English legal documents into Hindi. We believe that models trained on the MILDSum dataset can be very useful in future in making legal judgements accessible to the common masses in India.
>
> [G] As per our experiments over the MILDSum dataset, extractive methods showed superior performance compared to abstractive methods. In lines 310-313 of the paper, we already mentioned that the summaries in MILDSum have a mix of extractive and abstractive features, as they directly quote several lines from the original judgment text. Also, we have now computed coverage and density [https://aclanthology.org/N18-1065/] to measure the extractiveness /abstractiveness of the dataset. We find Extractive Fragment Coverage as 0.90, and Extractive Fragment Density as 24.42. So we get a high value for Coverage which measures the percentage of words in the summary that are part of an extractive fragment from the document/judgment. We also get a high value for Density which quantifies how well the word sequence of a summary can be described as a series of extractions. Since the summaries in our dataset contain directly quoted lines from original judgment, so Density is very high compared to other datasets (e.g., 3.94 for Arxiv/PubMed, 2.38 for BigPatent) [https://aclanthology.org/2022.findings-emnlp.488/]. This explains why Extractive summarizers outperform abstractive ones over our dataset. We will include this explanation in the final version.
>
> [H] We did not encounter any difficulties related to data privacy and security during the data collection process. The main challenge was for alignment between English and Hindi Summaries, which was reported in Section 3.2.
>
> Also, thank you for pointing out the typos and giving us suggestions regarding the improvement in Grammar Style and Presentation. We will definitely incorporate these in the final version (if accepted).

---

### Meta-Review · Area_Chair_Y6i5 · 2023-09-22

**Recommendation:** 5

**Metareview:**

This study introduces a multilingual legal document summarization dataset, comprising 3,122 judgment-summary pairs in both English and Hindi. In conjunction with the dataset, the authors presented benchmark results using various models.

All reviewers recognized the value of the dataset for this novel domain. Reviewer egHn raised several concerns, many of which have been addressed. However, the ethics statement remains unclear. To enhance clarity, the authors should elaborate (i) the implications of the models’ output, and (ii) the consent required for such data collection (as also mentioned by Reviewer eZYq). When releasing the dataset, it's imperative to ensure license terms accompany the released package. Concerning Q2 from Reviewer eZYq, providing a quantitative value would be more informative for readers.

Addressing Reviewer XjyQ's point on "some quantitative measures" is crucial to verify the dataset's quality. Although this concern was responded to in the rebuttal, such details should be incorporated into the paper.

---

### Decision · Program_Chairs · 2023-10-07

**Decision:**

Accept-Main

**Comment:**

This study introduces a multilingual legal document summarization dataset, comprising 3,122 judgment-summary pairs in both English and Hindi. In conjunction with the dataset, the authors presented benchmark results using various models.

All reviewers recognized the value of the dataset for this novel domain. Reviewer egHn raised several concerns, many of which have been addressed. However, the ethics statement remains unclear. To enhance clarity, the authors should elaborate (i) the implications of the models’ output, and (ii) the consent required for such data collection (as also mentioned by Reviewer eZYq). When releasing the dataset, it's imperative to ensure license terms accompany the released package. Concerning Q2 from Reviewer eZYq, providing a quantitative value would be more informative for readers.

Addressing Reviewer XjyQ's point on "some quantitative measures" is crucial to verify the dataset's quality. Although this concern was responded to in the rebuttal, such details should be incorporated into the paper.